# MFVC: Urban Traffic Scene Video Caption Based on Multimodal Fusion

**Mingxing Li** [1,2], **Hao Zhang** [1,2], **Cheng Xu** [1,2,*], **Chenyang Yan** [1,2], **Hongzhe Liu** [1,2] and **Xuewei Li** [1,2]

1    Beijing Key Laboratory of Information Service Engineering, Beijing Union University, Beijing 100101, China
2    Institute for Brain and Cognitive Sciences, Beijing Union University, Beijing 100101, China
*    Correspondence: xc-f4@163.com

**Abstract:** With the development of electronic technology, intelligent cars can gradually realize more complex artificial intelligence algorithms. The video caption algorithm is one of them. However, current video caption algorithms only consider single-visual information when applied to urban traffic scenes, which leads to the failure to generate accurate captions of complex sets. The multimodal fusion algorithm based on Transformer is one of the solutions to this problem. However, the existing algorithms have the difficulties of a low fusion performance and high computational complexity. We propose a new video caption Transformer-based model, the MFVC (Multimodal Fusion for Video Caption), to solve these issues. We introduce audio modal data and the attention bottleneck module to increase the available information to describe the generative model and improve the model effect with less operation costs through the attention bottleneck module. Finally, the experiment is conducted on the available datasets, MSR-VTT and MSVD. Meanwhile, to verify the effect of the model on the urban traffic scene, the experiment is carried out on the self-built traffic caption dataset BUUISE, and the evaluation index confirms the model. This model can achieve good results on both available datasets and urban traffic datasets and has excellent application prospects in the intelligent driving industry.

**Keywords:** video caption; traffic scene; multimodal fusion; attention

## 1. Introduction

Artificial intelligence's rapid development has gradually divided into different research fields, such as computer vision (CV), natural language processing (NLP), and so on. Video captioning (VC) is the combination of CV and NLP, and the recognition algorithm of CV recognizes videos. The video caption is output by the generator algorithm of NLP [1]. At present, the field of video captioning has a variety of application prospects. In the urban road scene, a video caption can report the vehicle driving environment and the interaction between objects in the traffic location through text in real time and can give an early warning of scary places. It is significant in assisting visually impaired people and assisting safe driving danger warnings. People's perception is complex and multimodal, and the application scenarios of most algorithms are also complicated and diverse. For example, in the autonomous driving method, the driving environment of crewless vehicles is very complex. It is difficult to accurately describe the video shot by the front camera of crewless vehicles only by relying on a single mode. Therefore, scientists propose to apply multimodal information to complex and diverse application scenarios. With the continuous development of deep learning in recent years, the application of multimodal information has also made remarkable achievements. However, using the complementarity between different modes while avoiding the redundancy between other methods is still a challenging problem in complex scenarios. In this paper, we present a new video caption generation model based on the attention bottleneck [2] to solve the problems of a low performance and high computational complexity of multimode fusion algorithms based on Transformer. Audio and image modes are used to model the video through attention

so that the model can thoroughly learn the feature information of the two methods and generate a more accurate natural language caption.

Based on the above analysis, this paper optimizes the video caption generation model based on Transformer as follows:

1. Use visual and audio modal information to obtain a more accurate caption by fully using the available information in the video.
2. The AB module is used to assist the fusion of multimodal information in improving the fusion performance of multimodal details as much as possible while keeping the computational complexity low.

## 2. Related Works

### 2.1. Video Caption

Research on video captioning can be divided into methods based on templates [3] and sequence to sequence [4]. Now, it is a sequence-to-sequence approach to modeling. The sequence-based method maps visual feature information and text feature information to the same vector space and learns its distribution to realize the sequence-to-sequence mapping. Venugopalan et al. [4] first used the convolutional neural network (CNN) to extract visual image features and then learn the distribution in the vector space through LSTM to generate word sequences. Silvio et al. [5] added Inception-Resnet-V2 to extract the motion features of the video and then used soft-attention LSTM (SA-LSTM) as the decoder. Kim et al. [6] used CNN to extract 2D features of video frames and then fed the extracted features to a differential neural computer (DNC) to learn the contextual information. The features of video frames are input into a DNC in chronological order, and its memory can store contextual information and fully use contextual information to generate captions.

With Transformer [7] achieving great success in many artificial intelligence fields, such as natural language processing, computer vision, audio processing, etc., more and more researchers have begun to use Transformer to solve problems, and now, a variety of Transformer variants have been proposed. The success of Transformer has also brought new ideas to the study of visual–text cross modals. Image caption and video caption have appeared in a Transformer-based model. Chen et al. [8] directly used CNN to extract the features of video frames and input them into the original Transformer and then output the distribution of the next word on the vocabulary through the Softmax layer. However, video caption is a multimodal learning problem, and video features have significant redundancy between different time steps. Therefore, Jin et al. [9] introduced the sparse border-aware strategy (SBS) into all multiple attention blocks in the encoder and decoder to reduce the redundancy in video representation based on this problem. However, coupling defects still exist in this method. Gao et al. [10] proposed a dual-layer decoupled Transformer model named D2 to solve this problem. By decoupling the spatial-temporal representation into the "first-spatial-then-temporal" paradigm, the whole model can be trained end to end by connecting the pre-training task with the downstream study. Gad et al. [11] proposed two real-time video caption methods based on Transformer and LSTM by integrating machine learning and the Internet of Things (IoT). The neural network is trained by reading many video caption pairs to restrict the caption to a subject–verb–object (SVO) template while replacing multiple lyrics with one word. This makes the model lightweight enough to be used on IoT platforms.

Most of the above methods use single-modal information as the input, ignoring the complementarity of modal information. Some methods use multiple-modal information as input. Still, the processing method of multimodal information is single, which improves the performance and dramatically increases the computational complexity of the model.

### 2.2. Multimodal Fusion

Multimodal fusion is one of the hot topics in modern artificial intelligence. Multimodal fusion is used in many fields to solve problems, especially in the area of video

understanding. Modality usually refers to the current mode of certain information, such as image information collected by eyes, audio information collected by ears, and point cloud information collected by lidar. Multimodal refers to the combination of multiple-modal information in a specific form. Different modal information represents different perspectives on things. As the information we see and hear has its characteristics, additional modal information is complementary to each other. If the complementarity between further modal information can be reasonably applied, rich characteristic information can be obtained. The standard modal information is visual, point cloud, text, audio, etc. Jin et al. [12] proposed the method of multimode fusion to describe the video content. Video, image, audio, dialogue and speech information, and category information are used in the encoder. Then, a fusion network is used for fusion and input to the decoder to obtain sentences. Ramanishka et al. [13] extended the literature [4] to take advantage of additional multimodal features and used motion features extracted by C3D, video tag information extracted by MSR-VTT, and audio features extracted by pyAudioAnalysis. Xu et al. [14] designed a new fusion module based on LSTM called the sub-sum fusion unit, which utilizes multi-level attention, namely the interaction of two-modal information at the time-series layer and modal information layer.

Since Transformer's great success, the attentional [15,16] has been widely used in machine vision and natural language processing and is now one of the most popular modules in artificial intelligence. At the same time, the scientists also found that the operation process of the attention takes the weighted sum between multiple vectors, which is very suitable for the fusion of multimodal information [17]. Zhang et al. [18] used pictures as additional features in machine translation, that is, image information and text information are input simultaneously into the Transformer decoder, and then text features and image features are integrated into the decoder through attention. Although this method achieves the interaction between different modes, it ignores the influence of single modes themselves. Xu et al. [19] encoded text and image features, used two Interactive memory networks to learn the feature information of a single mode, and then used multi-interactive attention to learn Interactive information between multiple modes. However, this method requires a fusion calculation for each layer, so the operation cost is very high.

The above multimode fusion methods generally have high computational costs and computational complexity. To reduce the computational cost and simultaneously take into account the influence of the single mode itself and the interaction between different methods of information, a new multimode fusion scheme applied to the field of video understanding is proposed in the literature [2]. The author uses the attention bottleneck (AB) module to encode the video and audio information with its attentional and then introduces the AB module to accept the information of the two modes to interact with each other through the cross attentional. Because only in the middle stage of the decoder the fusion between the different modal information is carried out, the operation cost is significantly reduced, and the influence of single-modal information on itself is calculated.

### 2.3. Video Understanding Algorithm in Urban Traffic Scenarios

Video-to-text cross-modal learning has been developed so far. It has a wide range of application values in many fields, such as video recommendation, video retrieval no longer limited to video labels, and helping people with visual impairment by transforming visual information into text information [20]. Inspired by this, it is proposed to be applied to the field of intelligent transportation to guarantee the warning of dangerous traffic information for people with hearing impairment and other disabilities. However, there are relatively few applications of video caption algorithms in urban traffic scenes, so it has good research space and prospects.

The video captioning algorithm not only expresses the current state of the road in real time with the text but can also be used for warning of dangerous scenes, such as pedestrians and battery cars crossing the road, and warnings of the danger of sudden lane changes of vehicles in front of hazardous situations can be made. They can also be expressed through text driving. The video caption of urban traffic scenes can also ensure the safety of assisted

driving for people with disabilities. However, the urban traffic scene is very complex and changeable, with many kinds of traffic elements and participants. Therefore, this paper will use the BUUISE dataset with a massive amount of video data of urban traffic scenes for training.

## 3. Video Caption Model

At present, many scholars have used multimodal information to conduct research on video caption algorithms [8,9,21], but most of them use the method of post-fusion for multimodal information fusion. The late fusion cannot fully utilize the complementarity between multimodal information and will accumulate redundancy. Therefore, this paper proposes a medium-term fusion video caption network structure to increase the utilization of complementary information and remove the influence of redundant information as much as possible. Firstly, visual features and audio features of video are extracted through the I3D [22] network and VGGish network and then input into the Transformer model with the attention bottleneck module. The model network learns the feature information of visual and audio by self-attention and then realizes the fusion of multimodal information by introducing the cross attention of attention bottleneck.

### 3.1. Multimodal Fusion for Video Caption (MFVC)

The model is divided into three parts: encoder, decoder, and generator. First, the decoder part takes a set of features $V = (v_1, v_2, \ldots, v_T)$ as input and maps $V$ to a series of intrinsic representations $Z = (z_1, z_2, \ldots, z_T)$. A caption $w_{\leqslant t} = (w_1, w_2 \ldots, w_t)$ is represented as $e_{\leqslant t} = (e_1, e_2, \ldots, e_t)$ by word embedding. The decoder then takes the output $Z$ of the encoder and a word embedding representation $e_{\leqslant t}$ as input, and the encoder generates $g_{\leqslant t} = (g_1, g_2, \ldots, g_t)$. The generator models the distribution $p(w_{t+1} \mid g_{\leqslant t})$ of the next word over the vocabulary by the $g$ output of the decoder, and greediness selects the next word by taking the word with the highest probability until a unique ending tag is sampled. Subtitles are initialized with a start tag. The encoder of this model is divided into two stages: early stage and late stage, respectively. The early stage is the decoder of self-attention, and the late stage is the decoder of cross attention. The structure diagram of the MFVC model is shown in Figure 1.

Figure 2a represents that in the early encoder of each layer, single-modal information is learned by pairwise self-attention of all hidden units in this layer. Figure 2b means that the AB module is used in the later encoder to restrict the flow of attention, and AB tokens in each layer can receive the information of two modal tokens as well as the information of AB tokens in the previous layer, thus showing the fusion of multimodal information.

### 3.2. Multimodal Encoder

The multimodal fusion model proposed in this paper is based on the Transformer structure, and the multimodal information fusion is carried out in the encoder. The encoder in this paper is divided into two parts, the early part and the late part, respectively, as shown in Figure 1. Firstly, the early part of the model sampled the long t-second video to obtain the F-frame image, then used I3D and VGGish to extract the feature of the video, and then spliced the visual and audio feature representations as a single sequence. Assuming that the feature extracted from the video frame is $V_{rgb}$ and the audio feature is $V_{spec}$, the final sequence can be expressed as follows.

$$V = \left[ V_{rgb} \| V_{spec} \right] \tag{1}$$

where $\|$ represents the splicing operation. The original Transformer is then used to process the above sequence: The encoder consists of $L$ layers. The first layer inputs a set of features $V$ and outputs the internal representation $z^l$, while the next layer takes the output of the previous layer as its input.

$$z^{l+1} = \text{Encoder}\left( z^l; \theta \right) \tag{2}$$

where $z^l$ is the concatenation of $z^l_{\text{rgb}}$, $z^l_{\text{spec}}$.

$$Z = \left[ Z_{\text{rgb}} \| Z_{\text{spec}} \right] \tag{3}$$

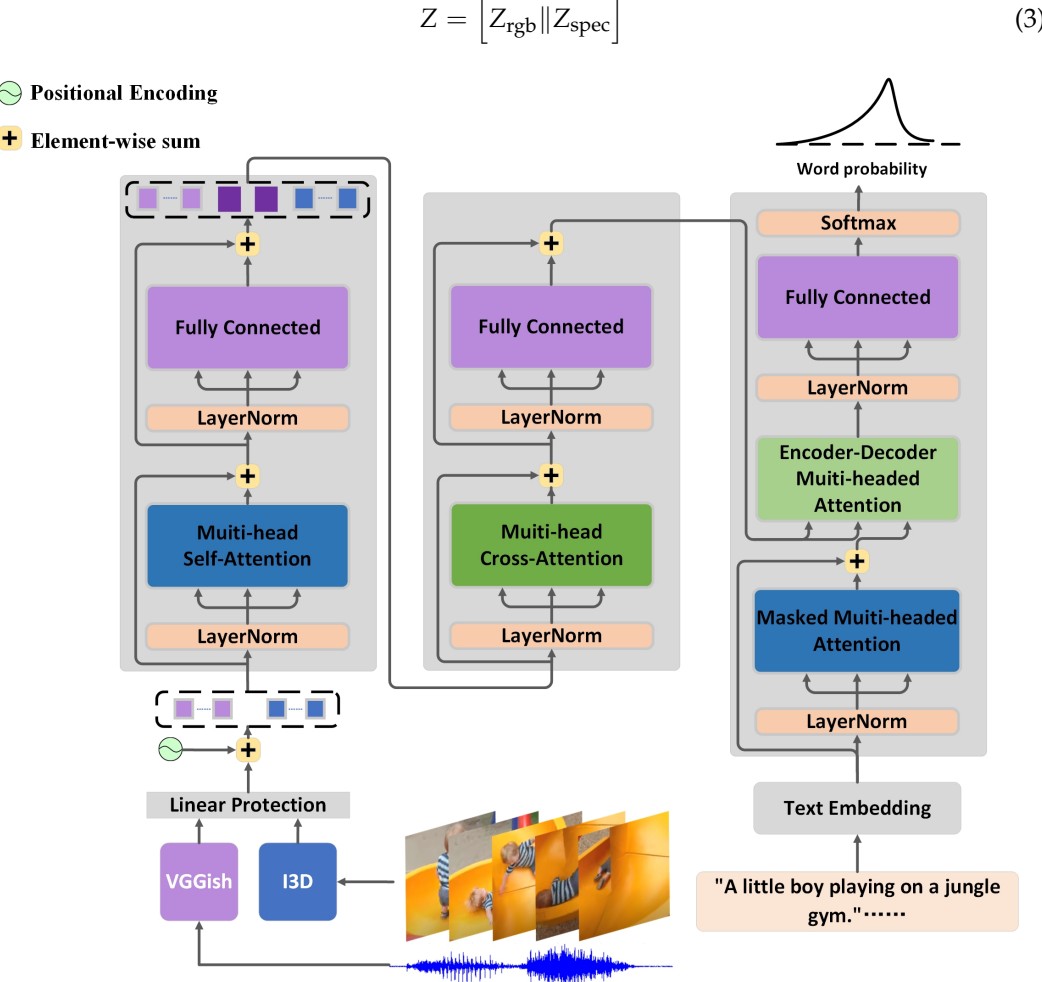

**Figure 1.** Video captioning generation using visual and audio modes can more accurately generate complex traffic scene videos. We use the attention bottleneck module for modal fusion.

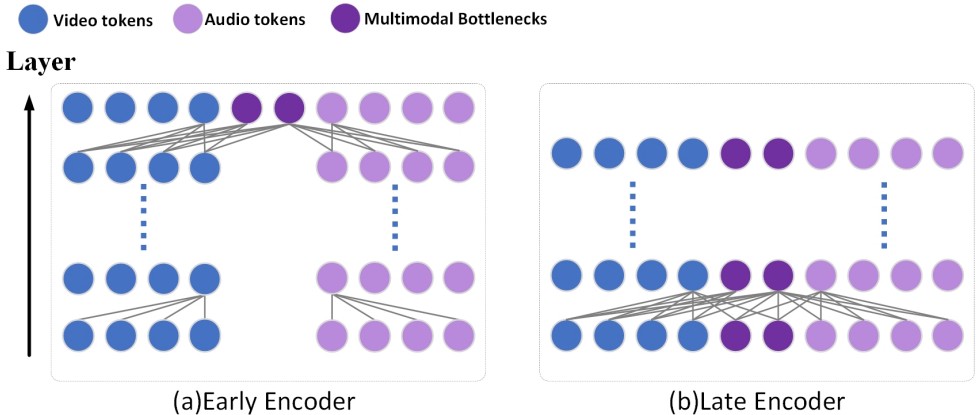

**Figure 2.** Each layer of tokens in the early encoders (**a**) and later encoders (**b**).

Each encoder layer $L$ consists of two sub-layers: a multi-head self-attention and a fully connected network. Layer normal is used to normalize the input of two sub-layers, and the input and output of each sub-layer are residual connected. Therefore, the $L$-layer encoder layer is defined as follows:

$$\bar{\mathbf{z}}^l = \text{LayerNorm}\left( \mathbf{z}^l \right) \tag{4}$$

$$\mathbf{r}^l = \mathbf{z}^l + \text{MSA}\left(\bar{\mathbf{z}}^l\right) \tag{5}$$

$$\bar{\mathbf{r}}^l = \text{LayerNorm}\left(\mathbf{r}^l\right) \tag{6}$$

$$\mathbf{z}^{l+1} = \mathbf{r}^l + \text{FCN}(\bar{\mathbf{r}}l) \tag{7}$$

The original Transformer encoder is shown in Figure 1, where the self-attention module can freely extract and process all the information from different modes. Multi-head self-attention (MSA) can be expressed as:

$$\text{MSA}(q,k,v) = \begin{bmatrix} \text{head}_1(q,k,v) \\ \dots \\ \text{head}_H(q,k,v) \end{bmatrix} W^o \tag{8}$$

$$\text{head}_h(q,k,v) = \text{Attention}\left(qW_h^q, kW_h^k, vW_h^v\right) \tag{9}$$

$$Attention(Q,K,V) = \text{softmax}\left(\frac{QK^T}{\sqrt{d_m}}\right)V \tag{10}$$

$W_h^q, W_h^k, W_h^v \in \mathbb{R}^{D_T \times D_k}$ for the training of projection matrix, the $q,k,v$ from $D_T$ mapping to $D_k = D_T/H$, $D_T$ is twice as large as the $H$. Meanwhile, the multi-head attention is the concatenated result of all attention heads mapping $D_T$ through the trainable parameter matrix $W^o \in \mathbb{R}^{D_k \cdot H \times D_T}$.

The output $A$ of the attention and the input $Z$ are connected by $A$ residual, namely $x = A + Z$. It is then input to the fully connected layer, and the calculation process of this operation is expressed as:

$$\text{FCN}(x) = \text{ReLU}(xW_1 + b_1)W_2 + b_2 \tag{11}$$

*3.3. Cross Encoder*

In this section, we set the $L = j$ layer as the first layer of the later encoder. In the later encoder, we replace the self-attention in the original Transformer with the cross attention for information exchange. The rest of the encoder is the same as the original Transformer. In addition, the attention bottleneck (AB) module is introduced into the input sequence as $\mathbf{Z}_{\text{fsn}}\left(z_{\text{fsn}}^1, z_{\text{fsn}}^2, \dots, z_{\text{fsn}}^B\right)$. Then, the input sequence of the later encoder becomes:

$$\mathbf{Z} = \left[\mathbf{Z}_{\text{rgb}} || \mathbf{Z}_{\text{fsn}} || \mathbf{Z}_{\text{spec}}\right] \tag{12}$$

In the first layer of cross encoder, we replay a tokens $\hat{\mathbf{Z}}_{\text{fsn}_i}$, $i$ is preferred *rgb,spec*.

$$\hat{\mathbf{z}}_{\text{fsn}_i}^j = \text{SelfAttention}\left(z_i^{j-1}; \theta_i\right) \tag{13}$$

The multi-headed cross attention (MCA) uses tensors $X$ and $Y$, where $X$ represents the query and $Y$ is used to re-weight the key and value of the query. It can be expressed as:

$$\text{MCA}(\mathbf{X}, \mathbf{Y}) = \text{Attention}\left(\mathbf{W}^Q\mathbf{X}, \mathbf{W}^K\mathbf{Y}, \mathbf{W}^V\mathbf{Y}\right) \tag{14}$$

Then, the cross-modal information is restricted to these AB modules. For each Transformer layer, the calculation process becomes:

$$\left[\mathbf{z}_i^{j+1} || \hat{\mathbf{z}}_{\text{fsn}_i}^{j+1}\right] = \text{CrossEncoder}\left(\left[\mathbf{z}_i^j || \mathbf{z}_{\text{fsn}}^j\right]; \theta_i\right) \tag{15}$$

$$\mathbf{z}_{\text{fsn}}^{j+1} = \text{Avg}_i\left(\hat{\mathbf{z}}_{\text{fsn}_i}^{j+1}; \theta_i\right) \tag{16}$$

where $\theta_i$ are the exclusive parameters of each module.

As a result, visual and audio feature vector updates can only be performed through AB modules, reducing computational complexity by limiting the number of AB modules to much smaller than the number of original tokens. Moreover, when cross-modal information is sent through fewer AB modules, the model forces each mode to condense its information, and only the essential information is sent to the other method, thus avoiding the transmission and calculation of redundant information in the method.

## 4. Experimental Results

The model validation is performed on two standard datasets, MSVD and MSR-VTT. To better verify the effect of the proposed method in urban traffic scenes, the dataset used in our experiments is the traffic scene dataset BUUISE. The following four evaluation indexes are used in this experiment: METEOR, BLEU-4 , Rouge-L, and CIDEr.

### 4.1. Datasets

MSVD is a public dataset released by Microsoft Research in 2010. The dataset consists of 1970 video clips, each containing an average of 40 human annotations.

MSR-VTT is a public dataset made public by Microsoft in 2016 for testing video caption models. The dataset consists of 10,000 video clips, each containing an average of 20 manual annotation statements. The English annotated statements in the above dataset were used for the model training.

BUUISE is a real-world road dataset built by the team from the Beijing Information Service Engineering Key Laboratory of the Beijing Union University. The dataset Buusie-Caption used in this paper is used to extract 3000 videos, and 6 to 20 s of each video is randomly captured as the original dataset and manually annotated. Each short video contains ten manual annotation statements.

### 4.2. Comparison with Other Methods

The MFVC model proposed in this paper is compared with other video caption models. Table 1 shows the results on the two datasets of MSVD and MSR-VTT. It can be seen from the results that the proposed method achieves the best results on each index in the two datasets. By comparing the TVT and SBAT algorithms, which also use I3D to extract video features, it can be seen that the multimodal fusion method in this paper has apparent advantages for the single mode. Table 2 shows the comparison of the parameter number and training time between the proposed model and other models, where V means video information and A means voice information. Compared with the TVT model, which also uses a Transformer structure, the model in this paper reduces the number of parameters and the training time of the model with the same two-modal information.

In Figure 3, the abscess coordinate represents the model training time, the ordinate coordinate represents the number of model encoder parameters, and the bubble area represents the CIDEr value. From this figure, it can be seen that the proposed method is very close to the number of parameters of TVT(BASE) using only a single mode, and it also has a better performance and shorter training time.

**Table 1.** Test results on MSVD, MSR-VTT test sets.

| | MSVD | | | | MSR-VTT | | | |
|---|---|---|---|---|---|---|---|---|
| **Methods** | **METEOR** | **BLEU@4** | **ROUGE-L** | **CIDEr** | **METEOR** | **BLEU@4** | **ROUGE-L** | **CIDEr** |
| SAAT [17] | 33.6 | 44.8 | 69.0 | 78.9 | 27.9 | 40.5 | 59.9 | 46.0 |
| TTA [23] | 33.8 | 50.6 | 70.4 | 83.0 | 27 | 39.6 | 60.2 | 45.7 |
| SGN [24] | 35.5 | 52.8 | 72.9 | 94.3 | 28.3 | 40.8 | 60.8 | 49.5 |
| Transformer-LST [25] | — | — | — | — | 27.9 | 38.6 | 60.2 | 44.6 |
| TVT [8] | 34.5 | 53.0 | 70.8 | 77.7 | 27.5 | 39.0 | 59.3 | 45.7 |
| Vanilia T [8] | 34.6 | 51.4 | 69.7 | 86.4 | 28.5 | 40.9 | 60.4 | 48.9 |
| SBAT [9] | 35.3 | 53.1 | 72.3 | 89.5 | 28.9 | 42.9 | 61.5 | 51.6 |
| Ours | 36.7 | 54.6 | 73.6 | 90.4 | 29.6 | 43.8 | 63.2 | 52.5 |

**Table 2.** Performance comparison of MSR-VTT datasets.

| **Methods** | **Types** | **Params** | **Training Time (s)** | **METEOR** | **BLEU@4** | **ROUGE-L** | **CIDEr** |
|---|---|---|---|---|---|---|---|
| BiLSTM [8] | BiLSTM(V) | 27M | 2291 | 26.34 | 36.96 | 58.71 | 43.09 |
| TVT(base)[8] | Transformer(V) | 25M | 805 | 26.88 | 37.26 | 58.31 | 44.57 |
| Att-tvt [8] | Transformer(V+A) | 30M | 1483 | 28.24 | 42.46 | 61.07 | 48.53 |
| Ours | Transformer(V+A) | 25M | 1082 | 29.62 | 43.84 | 63.23 | 52.54 |

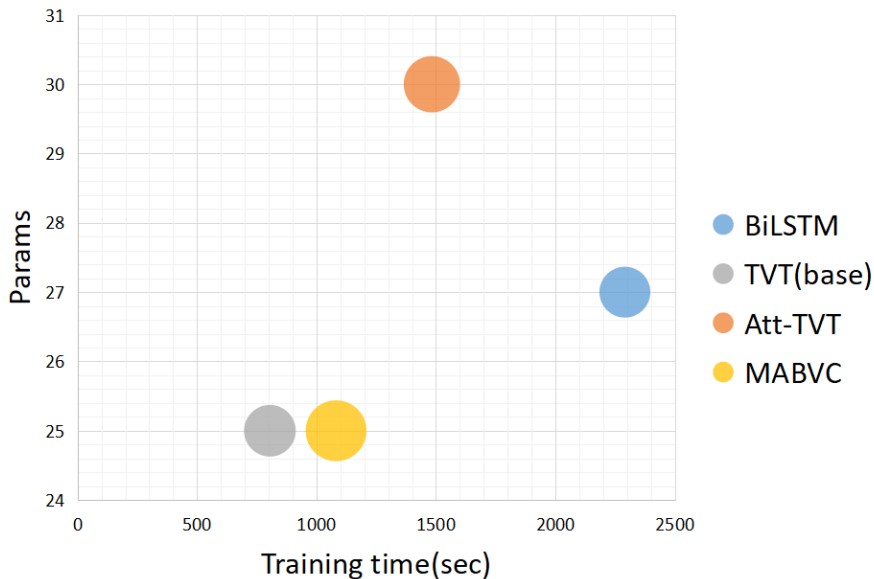

**Figure 3.** Comparison of parameters. Training time and CIDEr values of different methods in MSR-VTT dataset.

The video caption of the proposed method on the BUUSIE dataset is shown in Figure 4, showing the manual annotation statements of some data and the statements generated by the proposed method. You can see that the model automatically generated by the comments can contain wealthy language elements and adapt to more complex content scenarios. This is due to the more diverse model input, which is more in line with people's habit of describing the video, so the model produces better results.

| | GT | Vanilia T | SBAT | Our |
|---|---|---|---|---|
| | A white car is pulling up ahead. | A black car is driving towards a white car. | A black car is crossing the zebra crossing. | A black car is turning right on our right. |
| | A black car is turning right. | | | |
| | An e-bike is driving on the right. | A bicycle is crossing the road on the right | A white car was parked by the side of the road. | A bike is driving on the right and crossing the road. |
| | An e-bike is crossing the road. | | | |

**Figure 4.** Video caption instance in BUUISE dataset [8,9].

Table 3 shows the comparison between the proposed method and other open-source algorithms on the BUUSIE dataset. It can be seen that the method using multimodal information for training in more complex scenes is better than the single-modal methods.

**Table 3.** Test results on BUUSIE test sets.

| Methods | METEOR | BLEU@4 | ROUGE-L | CIDEr |
|---|---|---|---|---|
| TVT [8] | 26.84 | 39.35 | 59.2 | 73.25 |
| Vanilia T [8] | 27.32 | 41.24 | 61.94 | 86.7 |
| SBAT [17] | 29.54 | 45.61 | 61.8 | 87.13 |
| Our | 31.36 | 51.42 | 65.62 | 90.2 |

*4.3. Ablation Experiment Analysis*

This paper uses the AB module to improve the Transformer network. All the indicators are improved compared with SBAT and TVT, which also use multimodal methods. The ablation experiment was performed on two datasets, MSVD and MSR-VTT, and the effect of the AB module was reflected in the experimental results.

It can be seen from Table 4 that when the multimodal information is used, the results of the BASE model are different from those of the single mode because only self-attention is used. There is no information interaction between the multimodal information, and redundant information exists between the two methods. However, the MFVC model introduces cross attention, and the information of multiple modes can interact with each other. The result is better than the single mode and the ATT-tvt model, which also uses video and audio modes. However, the AB module is introduced, and even if the self-attention is used only, the information between different methods can still be sent through the AB module, so the results are improved. The self-attention and cross attention achieved the best results after using the AB module.

**Table 4.** Ablation test results on MSVD and MSR-VTT test sets.

| Methods | AB | MSVD | | | | MSR-VTT | | | |
|---|---|---|---|---|---|---|---|---|---|
| | | M | B@4 | R-L | C | M | B@4 | R-L | C |
| TVT [8] | N/A | 34.5 | 53.0 | 70.8 | 77.7 | 27.52 | 38.96 | 59.33 | 45.67 |
| Vanilia T [8] | N/A | 34.6 | 51.4 | 69.7 | 86.4 | 28.5 | 40.9 | 60.4 | 48.9 |
| Base(MSA) | N/A | 34.2 | 51.1 | 69.4 | 83.6 | 26.2 | 38.2 | 58.2 | 43.6 |
| Base(MSA) | 2 | 34.6 | 52.3 | 70.8 | 84.3 | 28.6 | 42.1 | 69.4 | 44.3 |
| MFVC(Mix) | N/A | 35.2 | 53.4 | 72.1 | 89.6 | 29.2 | 43.4 | 62.1 | 49.6 |
| MFVC(Mix) | 2 | 36.7 | 54.6 | 73.6 | 90.4 | 29.6 | 43.8 | 63.2 | 52.5 |

We conducted experiments on the impact of the AB module number *B* in different fusion layers *L*. We investigated the models with different AB modules and fusion layers and present the results in Figure 5. From the results, the AB modules improve the performance at a lower computational cost.

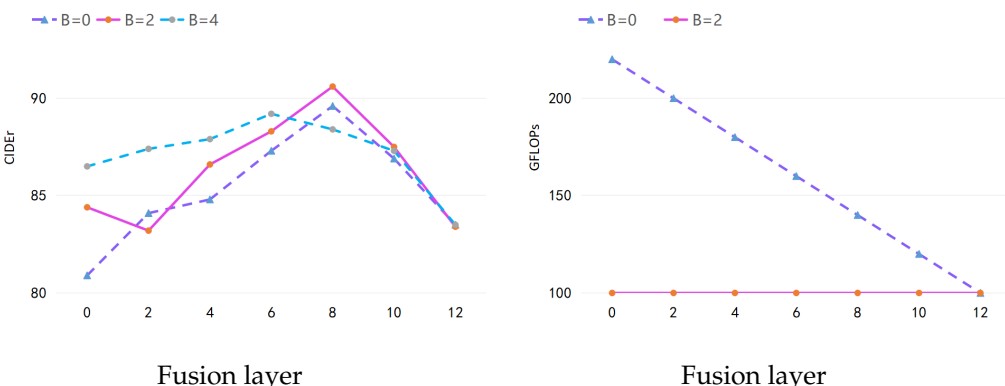

**Figure 5.** The impact of using the different numbers of AB modules (B is the number of AB modules) for fusion on performance (**left**) and compute (**right**) at different fusion layers on the dataset.

## 5. Discussion

In this section, we will analyze the experimental data and conduct a quantitative analysis of the experimental data to show the advantages of our method in terms of parameters and multimodal information integration.

The experimental results in Tables 1 and 2 show that MFVC has the lowest parameter number and training time while maintaining a high performance on the two datasets. MFVC's overall performance is better than Att-tvt and SBAT [8,9], which use multimodal information and Transformer architecture. Att-tvt uses dual-view features to fuse at the decoder to increase the video's visual representation and audio information to improve the available data of the model. SBAT uses 2D and 3D features with similar features for multimodal fusion and reduces the redundancy between video features by pooling the scores of multi-head attention.

According to the experimental results in Table 3, the performance of MFCV on the traffic road dataset BUUSIE is better than the other methods. Audio information can help the model generate more accurate captions in complex traffic road scenes. Compared with other multimodal methods, MFVC has a higher degree of multimodal information fusion, and BUUISE has sufficient data to provide MFVC for training, thus achieving a much higher performance than other methods.

## 6. Conclusions

This paper proposes a new multimodal fusion video caption method based on Transformer. The experiments show that video information and audio information can be used to achieve complementary information to make a more accurate text caption. The advantage of this network is that it can better use the helpful information between multiple modes. The experimental verification on the MSVD and MSR-VTT datasets shows that the proposed method achieves promising results on the commonly used evaluation indicators METEOR, BLEU-4, Rouge-L, and CIDEr. It also achieves good results on the traffic scene dataset BUUSIE. It is proved that this method applies to traffic road scenarios and has good generalization. This algorithm cannot perform well on small datasets, but our model can achieve a competitive performance on the MSVD, MSR-VTT, and BUUISE datasets. We hope to improve the performance of small datasets with more lightweight models in the future.

**Author Contributions:** Conceptualization, M.L.; methodology, M.L.; software, M.L. and H.Z; validation, M.L., C.Y., H.Z. and C.X.; formal analysis, C.X.; investigation, M.L. and H.Z.; writing—original draft preparation, M.L. and H.Z.; writing—review and editing, M.L., H.L. and X.L.; visualization, M.L.; supervision, H.L. and X.L.; project administration, M.L. All authors have read and agreed to the published version of the manuscript.

**Funding:** This work was supported by the National Natural Science Foundation of China (Grant No. 62171042, 62102033, 61871039, 62006020, 61906017), the R&D Program of Beijing Municipal Education Commission (KZ202211417048), the Beijing Municipal Commission of Education Project (No. KM202111417001, KM201911417001), the Collaborative Innovation Center of Chaoyang (Grant No. CYXC2203), and the Academic Research Projects of Beijing Union University (No. BPHR2020DZ02, ZK40202101, ZK120202104).

**Data Availability Statement:** Not applicable.

**Conflicts of Interest:** The authors declare no conflict of interest.

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
