# Peer review of "MFVC: Urban Traffic Scene Video Caption Based on Multimodal Fusion"

_electronics, doi:10.3390/electronics11192999_

Round 1
Reviewer 1 Report
This study proposed a new video caption model MFVC based on the Transformer by introducing audio modal data and attention bottleneck module to improve the model effect with less operation cost through the attention bottleneck module. Comparative experiments were performed on general datasets MSR-VTT, MSVD, and self-built dataset BUUISE. The result showed that the model can obtain good results on all the datasets.
The findings reported in this study are quite refreshing. I would like to state that the study had been meticulously conducted and the research gaps had been pointed out and filled in. There are still some issues to be addressed prior to the acceptance.
1. The following statement will require some citations to support. "At the same time, the scientists also found that the operation process of the attention takes the weighted sum between multiple vectors, which is very suitable for the fusion of multi-modal information."
2. The final section should be "Conclusions". The authors should discuss their results with previous findings in the section of discussion.
3. A section of "Limitations of the Study" or "Suggestions to the Future Studies " would be suggested.
Author Response
Response to Reviewer 1 Comments
Point 1: The following statement will require some citations to support. "At the same time, the scientists also found that the operation process of the attention takes the weighted sum between multiple vectors, which is very suitable for the fusion of multi-modal information."
Response 1: I have added the reference to the part you proposed as requested. The additions is highlighted in the pdf file.
Point 2: The final section should be "Conclusions". The authors should discuss their results with previous findings in the section of discussion.
is
Response 2: The conclusion section has been added as you requested, and the content of the discussion section has been added. The additions is highlighted in the pdf file.
Point 3: A section of "Limitations of the Study" or "Suggestions to the Future Studies " would be suggested.
Response 3: I have supplemented the conclusion according to your suggestions. The specific contents refer to the two aspects of your suggestion. And the additions is highlighted in the pdf file.

Reviewer 2 Report
In the present work, the authors tried to model a video caption algorithm to overcome the single information algorithms for urban traffic scenes. For this, the authors employed a multi-modal fusion algorithm based on the Transformer of audio modal data and attention bottleneck module to increase the available information captured. The outcomes then are revealed and compared to other models. After reviewing the manuscript, the following are my comments.
- The paper is thorough, well written, has a strong literature review, a clear description of the methodology, and makes its contribution clear to the reader by significant results. However, the Authors should also proofread these parts of the paper for minor grammatical and spelling errors.
- The second part: Discussion and conclusion sections need major reconstruction and presentation; the authors did not discuss the results sufficiently, and greater clarification of models comparison results is needed. Moreover, a section of conclusions that highlights the key findings and points out the limitations of the study is missed.
Author Response
Response to Reviewer 2 Comments
Point 1: The paper is thorough, well written, has a strong literature review, a clear description of the methodology, and makes its contribution clear to the reader by significant results. However, the Authors should also proofread these parts of the paper for minor grammatical and spelling errors.
Response 1: The full text has been reviewed per your request, and several grammatical and spelling corrections have been made. The additions is highlighted in the text
Point 2: The second part: Discussion and conclusion sections need major reconstruction and presentation; the authors did not discuss the results sufficiently, and greater clarification of models comparison results is needed. Moreover, a section of conclusions that highlights the key findings and points out the limitations of the study is missed.
Response 2: The discussion and conclusion sections have been revised and supplemented as requested. Additions include further clarification of the results of model comparisons and pointing out study limitations. The additions is highlighted in the text

Round 2
Reviewer 2 Report
According to the modified version of the manuscript, The reviewer thinks it's now ready for publishing.